# Comparative analysis of pulmonary ventilation distribution between low-cost and branded incentive spirometers using electrical impedance tomography in healthy adults: Study protocol

**Noraelena Mera, Esther Cecilia Wilches Luna, Vicente Benavides-Cordoba** *

Universidad del Valle, Cali, Colombia

* vicente.benavides@correounivalle.edu.co

## Abstract

### Background

The Incentive Spirometer (IS) increases lung volume and improves gas exchange by visually stimulating patients to take slow, deep breaths. It prevents respiratory complications and treats postoperative atelectasis in patients undergoing abdominal, thoracic, and neurosurgical procedures. Its effectiveness has been validated in studies that support improved lung capacities and volumes in individuals with respiratory complications, postoperative thoracic surgery, upper abdominal surgery, and bariatric surgery. The modified Pachón incentive spirometer (MPIS) is a cost-effective alternative to branded IS. It is crucial to validate whether the MPIS distributes ventilation as effectively as commercial devices do. Ventilation distribution will be measured using electrical impedance tomography.

### Objective

The aim is to compare the distribution of pulmonary ventilation between the MPIS and another commercial IS in healthy adults using electrical impedance tomography.

### Methods

A crossover clinical trial is proposed to evaluate the measurement of pulmonary ventilation distribution using EIT in a sample of healthy adults. All participants will use a commercial flow IS and the MPIS, with the order of assignment randomized. This research will use electrical impedance tomography to validate the operation of the MPIS.

### Conclusions

This study protocol will compare two incentive spirometers' impact on pulmonary ventilation, potentially endorsing the adoption of a cost-effective device to enhance accessibility for targeted populations.

relevant data from this study will be made available upon study completion.

**Funding:** The author(s) received no specific funding for this work.

**Competing interests:** The authors have declared that no competing interests exist.

## Trial registration

The study was registered in ClinicalTrials.gov (NTC05532748).

## Introduction

The Incentive Spirometer (IS) increases lung volume and improves gas exchange [1] by visually stimulating patients to take slow, deep breaths [2]. It prevents respiratory complications and treats postoperative atelectasis in patients undergoing abdominal, thoracic, and neurosurgical procedures [3]. Its effectiveness has been validated in studies that support improved lung capacities and volumes in individuals with respiratory complications, postoperative thoracic surgery, upper abdominal surgery, and bariatric surgery [4,5]. Once the patient is trained, supervision is not required to use it, thus reducing the amount of time the patient has direct contact with the therapist [4].

Several commercial IS have been developed, including Voldyne®, Triflo®, Inspirx®, RSB®, and SpiroBall®, among others [2]. However, the prices of these devices can be prohibitive for many users. Patients may be unable to access these devices' benefits due to limited resources or lack of availability in their institutions. Limited access to assistive devices is a social and public health issue that affects Latin American cities and countries, preventing those who could benefit from their use [6]. Therefore, cost-reducing initiatives are constantly being sought. For instance, a Colombian anesthesiologist, Mauricio Pachón, built an incentive device using recyclable, low-cost, and easy-to-manufacture materials. This device has been modified and characterized as a flow IS, known as the Modified Pachón Incentive Spirometer (MPIS) [7].

In the clinical setting, evaluation forms commonly used to monitor the effectiveness of lung expansion therapy are based on global parameters such as oxygen saturation, arterial blood gas measurements, and semiological and radiological assessments [8]. However, these global parameters do not provide sufficient sensitivity to objectively assess the regional redistribution of ventilation in response to lung expansion therapy. In addition, it is essential to note that chest radiography and computed tomography have limitations. Chest radiography is limited by its static component, while computed axial tomography is expensive and exposes patients to radiation [9].

The aforementioned has led to the need for alternative measurement forms, such as Electrical Impedance Tomography (EIT). EIT utilizes the electrical properties of tissue to obtain non-invasive and continuous information at the bedside without radiation [10]. This technique allows for monitoring ventilation and pulmonary perfusion by providing repeated images of tidal volume distribution It distinguishes each lung's regional filling and emptying characteristics in real-time and with safety. The significance of safe and sustainable interventions also relies on the cost of the devices, particularly in regions with limited resources.

This study aims to compare the distribution of ventilation between low-cost and branded devices using EIT measurements. The EIT measurement provides real-time data in different regions and moments that traditional tests cannot obtain. This could lead to further studies in patients with restrictive changes. This research will use EIT to validate the operation of the MPIS.

The aim is to compare the distribution of pulmonary ventilation between the MPIS and another commercial IS in healthy adults using electrical impedance tomography.

## Methods

### 2.1 Study setting

All measurements for this trial will be performed at the measurement and evaluation laboratory of the School of Human Rehabilitation at the Universidad del Valle in Cali, Colombia. The study was registered in ClinicalTrials.gov (NTC05532748).

### 2.2 Trial design

A crossover clinical trial is proposed to evaluate the measurement of pulmonary ventilation distribution using EIT in a sample of healthy adults. All participants will use a commercial flow IS and the MPIS, with the order of assignment randomized. The trial will be performed following the recommendations of the Consolidate Standards of Reporting Trials (CONSORT) [11]. In addition, the SPIRIT guidelines [12] have been followed. (Fig 1).

### 2.3 Participants, eligibility criteria, and recruitment

Adults between the ages of 18 and 65 who are in good health will be included. To prevent bias, eligibility criteria were used to screen participants for health conditions that could affect the study. Individuals meeting the following criteria will be included: clinical stability, defined as the absence of any acute illness in the previous 6 weeks; Charlson index classification of 0–1; body mass index between 18.5 to 35 kg/m2; and no cognitive alterations. Patients with pacemakers, cardioverters, metal implants, obstructive or restrictive lung function disorders, or a high level of physical activity according to the IPAQ short version questionnaire will be excluded from consideration.

| | STUDY PERIOD | | | | | |
|---|---|---|---|---|---|---|
| | Enrolment | Allocation | Post-allocation | | | Close-out |
| TIMEPOINT** | $-t_1$ | $t_0$ | $t_1$ | $t_2$ | $t_3$ | $t_x$ |
| **ENROLMENT:** | | | | | | |
| Eligibility screen | X | | | | | |
| Informed consent | X | | | | | |
| Allocation | | X | | | | |
| **INTERVENTIONS:** | | | | | | |
| Intervention MPI | | | X | X | X | |
| Intervention Triflo II | | | X | X | X | |
| **ASSESSMENTS:** | | | | | | |
| **Baseline variables** | | | | | | |
| Age | X | | | | | |
| Gender | X | | | | | |
| BMI | X | | | | | |
| IPAQ (Short form) | X | | | | | |
| Charlson Index | X | | | | | |
| Spirometry (FEV1, FVC, FEV1/FVC) | X | | | | | |
| **Outcome variables** | | | | | | |
| VTM | | | X | X | X | |
| MV | | | X | X | X | |
| ΔEELI | | | X | X | X | |
| **Other data variables** | | | | | | |
| Heart rate | | | X | X | X | X |
| Breathing Frequency | | | X | X | X | X |
| Oxygen Saturation | | | X | X | X | X |
| Borg Scale | | | X | X | X | X |
| Customer satisfaction survey | | | | | | X |

**Fig 1. SPIRIT schedule of enrolment, interventions, and assessments.** MPI: Modified Pachón incentive, BMI: Body mass index, IPAQ: International physical activity questionnaire, FEV1: Forced expiratory volume in the first second, FVC: Forced vital capacity, ΔEELI: Delta end-expiratory lung impedance, VT: Tidal variation, MV: Minute variation.

**2.3.1 Recruitment.** Volunteers will be recruited through communication channels established by the Universidad del Valle. These channels include social networks, emails, and word of mouth to create the snowball effect. The call will provide participants with all the necessary information before contacting the researchers. Individuals who choose to enroll in the study will receive communication regarding the date, time, and location of the study. These details will be arranged based on availability and the capacity and biosafety protocols of the study site. It is crucial to adhere to these measures.

Spirometry will be performed using the Medgraphics Cardiorespiratory Diagnostics® spirometer to confirm normal lung function in the volunteer. The test will be conducted under an established biosafety protocol to ensure the safety of all participants, witnesses, and researchers. The spirometry will adhere to the American Thoracic Society and the European Respiratory Society standards [13]. Participants must abstain from physical activity, smoking, vaping, or consuming caffeinated beverages for at least 12 hours before the exam. The pulmonary function laboratory at Hospital Universitario del Valle will conduct the measurement, with the test technician being blinded to the IS assigned to each participant. After completing spirometry, participants will be randomized to determine their initial incentive and who will undergo the first ventilation measurement using EIT.

## 2.4 Interventions

**2.4.1 Incentive spirometers.** The Modified Pachon IS is a device that imitates conventional respiratory incentives. It is constructed using recyclable, low-cost materials and is easy to manufacture. The device is made up of a buretrol and an unlubricated Today® brand condom. It is essential to avoid using condoms with lubricant as they can cause discomfort during handling, leading to the condom sticking to the walls of the buretrol and hindering its displacement. The MPIS is a flow incentive that mobilizes flows ranging from 600 cc/sec to more than 1400 cc/sec. The Colombian Superintendence of Industry and Commerce has endorsed its industrial design. These devices will be mass-produced by a mechanical engineer and with sterile materials.

Triflo® II IS is a method used to promote deep voluntary breathing. It provides visual feedback on inspiratory volume through three color-coded balls in three chambers, a mouthpiece, and a tube. The minimum flow is marked in each chamber: 600, 900, and 1200 ml/Sec. It is used for inspiratory muscle training and prolonged maximal inspiration. Deep breathing stimulates the alveoli to expand fully. HUDSON RCI® manufactures this device often used to prevent or reverse collapsed lung formation. This device is used in clinical practice and related research.

**2.4.2 Measurements.** The study employs a crossover design, in which subjects receive two or more procedures at different periods, with the sequence of treatments randomized for each subject. According to the data collected during the pilot test, each procedure is expected to take approximately 30 minutes. Measurements are taken twice, with the first measurement using the randomly assigned incentive and the second measurement occurring one week later using the alternative incentive to ensure a thorough washout period. The principal investigator conducts both measurements. Throughout the washout period, to minimize the risk of sample loss, two reminder calls will be scheduled for the second measurement, while participants will be instructed to refrain from conducting lung re-expansion maneuvers during this period.

The study will assess the distribution of ventilation induced by each Incentive Spirometer using electrical impedance tomography equipment, specifically the PulmoVista500® from Drägerwerk AG & Co. KGaA. The procedure requires the patient to be seated in a high-back chair, with the belt always resting on a surface and the equipment signal being high. A belt

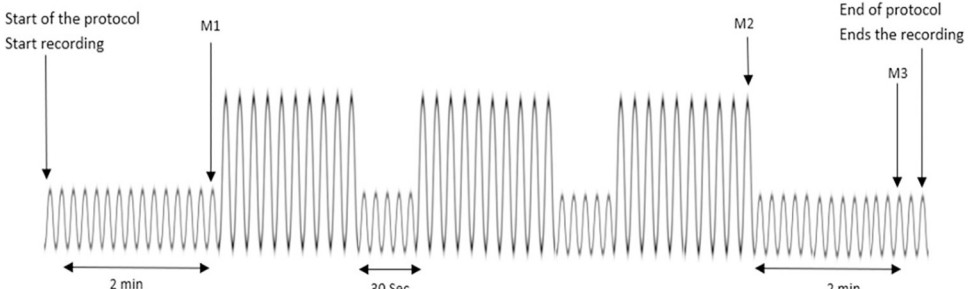

**Fig 2. Illustration of reexpansion protocol and each measurement moment.** M1: Moment 1, M2: Moment 2, M3: Moment 3.

with 16 integrated electrodes is placed around the patient's chest between the fifth and sixth intercostal space. The belt was selected based on the participant's chest diameter to ensure a proper fit. A video demonstration will provide instructions on using the corresponding incentive, illustrating the correct technique for performing re-expansion therapy. Although the frequency and duration of exercises are still debated, current recommendations suggest three sets of ten repetitions, with rest intervals of 30 to 60 seconds between each set [14]. In this case, three sets of ten breaths will be performed using the IR technique, with a 30-second rest between sets.

The patient is instructed to inhale gently and deeply through their mouth, hold for 5 seconds, and exhale slowly through their mouth without removing the mouthpiece from the incentive, exerting no additional effort. The device will provide visual feedback indicating either the elevation of the condom inside the buretrol for the MPIS or the raising of spheres in the commercial branded incentive. The measurement with electrical impedance tomography will begin once the patient is prepared.

The study will begin with two minutes of relaxed breathing, followed by the marking of the first event, referred to as Moment 1 (M1). Subsequently, the lung re-expansion protocol will be executed, which involves three sets of ten breaths with a 30-second rest between sets. After the final repetition of the last set, the second event will be marked as Moment 2 (M2). Finally, the participant will engage in two minutes of calm breathing, and the third event corresponding to Moment 3 (M3) will be marked (Fig 2).

Clinical data, including heart rate, respiratory rate, and oxygen saturation, will be recorded at the beginning, end, and 15 minutes after the re-expansion maneuver. The study will evaluate participant satisfaction with the MPIS device, which is not commonly used daily. This evaluation will be conducted through a Likert rating scale survey. The survey will include questions about the design, materials, colors, and comfort of the MPIS during use. Additionally, the study will assess participants' perception of dyspnea using the Borg scale. After completing each measurement with every participant, the equipment will be disinfected according to the manufacturer's recommendations.

The session may be interrupted under specific circumstances. These include if the participant chooses not to continue with the re-expansion technique, if the maneuver causes intolerance, if the respiratory rate exceeds 25 breaths per minute, or if accessory muscles are utilized. In the event of adverse reactions, procedures will be immediately suspended, documented in the database, and promptly reported to the relevant institutional authorities.

## 2.5 Outcomes (primary and secondary)

**2.5.1 Primary outcomes measure.** Delta EELI (ΔEELI) is a measure of changes in end-expiratory lung impedance before and after an intervention. This can be interpreted as changes

in end-expiratory lung volume in the electrodes' plane. It represents the functional residual capacity and is directly related to lung recruitment.

Tidal Variation (TV)is the sum of the regional relative impedance changes of the entire state image or within the defined region of interest (ROI). It is related to the increased resistance of the lung tissue to the passage of current, which increases with the entry of air and decreases with its exit.

Minute tidal variation (MTV) is the average of the sum of the global or regional relative impedance changes in the last minute of the entire state image or within the defined region of interest. It is related to the increased resistance of the lung tissue to the passage of current in the last minute, which increases with the entry of air and decreases with its exit.

**2.5.2 Secondary outcomes measures.** IPAQ: Instrument that provides information on estimated energy expenditure in 24 hours in different areas of daily life.

Charlson Index: Method that relates long-term mortality to participant comorbidity.

%FEV1 Pred: Percentage of predicted forced expiratory volume in the first second.

%FVC Pred: Percentage of predicted forced vital capacity.

%FVC/FEV1: Percentage of the predicted relationship between forced vital capacity and FEV1.

## 2.6 Sample size

The study will include 30 healthy volunteers from Santiago de Cali, Colombia, with equal representation of both sexes (15 men and 15 women). Convenience sampling will be used. As a pilot trial, the objectives differ from those of a main trial, making formal power considerations for determining sample size typically unnecessary [15,16]. This study aims to explore the initial behavior of a new respiratory device, the MPIS, regarding lung ventilation distribution. This study will compare the device with another commercially available respiratory device. This research falls under the category of a phase 0 clinical trial, which is characterized by the use of small sample sizes, often with fewer than 15 participants, and administering the drug and/or treatment for a brief period [17]. Reviewers also recommend small samples for this type of study during the registration phase on the clinicaltrials.gov website.

## 2.7 Randomization

A simple randomization approach will allocate participants to the sequences, determining the initial incentive assigned to each participant and the individual selected for the initial ventilation measurement. This method ensures unbiased assignment, where each participant has an equal chance of being allocated. By utilizing simple randomization, the study aims to maintain group comparability and enhance the robustness of the findings. To achieve unbiased randomization of participants, we will use http://www.randomization.com and dedicated statistical software that employs random number generation. This will result in a balanced distribution of participants across both treatment sequences. A concealment procedure will ensure that researchers and participants are blinded to the assigned treatment group. All the randomization process, will be conducted by an impartial researcher not involved in data collection.

## 2.8 Blinding

For the main researchers and participants, the allocation process for the first incentive and the subsequent statistical analysis will be conducted in a blinded manner. However, due to the nature of the intervention, researchers and participants cannot be blinded as they are aware of the device they use during the maneuver.

## 2.9 Data collection and management

The collection of real-time data will be safeguarded by coding the participants to ensure confidentiality. To maintain the integrity of the recorded information, random checks will be conducted to validate consistency among the original team data, recorded forms, and database entries. Any identified errors will be promptly rectified.

As for data stored in the tomograph, monitoring sessions will be archived alongside the participant's code, securely stored in the equipment for future reference if necessary.

A comprehensive checklist will be developed to ensure each participant adheres meticulously to the research protocol, thereby preventing the omission of pertinent information that could introduce bias or lead to data loss. In instances where missing data cannot be retrieved, the participation of the subject will be excluded from analysis.

## 2.10 Statistical analysis

The data as study variables will be coded, processed and analyzed, and will be recorded using the designed data collection formats and later organized in Microsoft Excel®. This will provide an initial understanding of the behaviors of each variable. If the distribution is normal, quantitative variables will be presented as means and standard deviations. If not, the median will be estimated using positional measurements. A normality test (Kolmogorov-Smirnov) will be applied to identify the normal distribution of the variables. For qualitive variables, frequencies and percentages will be estimated. A descriptive bivariate analysis will compare the distribution of global and regional pulmonary ventilation according to sociodemographic and anthropometric characteristics. A T-test will be used for related samples if the variables have a normal distribution to determine if changes after interventions are statistically significant. Otherwise, the non-parametric Wilcoxon test will be used with a significance level of 5%. The statistical procedure will be conducted using SPSS Statistics 21.

## 2.11 Quality assurance

To ensure the accuracy of recorded information, an external researcher who is not involved in the measurement process, will conduct random reviews to verify the consistency between the original equipment information, the recorded formats, and the database. Any errors detected will be corrected. Random reviews of the forms will be conducted every 15 days to verify the accuracy of the information before it is entered into the database. Additionally, there will be a monthly meeting with the research group to discuss the progress of the project.

## 2.12 Ethics

The Universidad del Valle Ethics Committee approved the project with code 009–022 of May 2022. If adjustments or changes to the protocol are necessary, a project amendment will be requested from the committee. Before enrollment, patients will be required to sign the informed consent form and agree to participate in the study. The data collected during the clinical trial will be deposited on clinicaltrials.gov and figshare.com. Following acceptance for publication, the data will be made accessible for public viewing.

# Discussion

This research aims to significantly contribute to the progress of research and innovation in respiratory physiotherapy. It seeks to achieve this goal by conducting a study that compares a low-cost IS (approximately $2.5 USD) with a branded flow incentive (approximately $8 USD), ensuring the reliability of the former. In this way, the study aims to improve accessibility and

benefit vulnerable populations who face challenges in acquiring commercial incentives. It also aims to provide an alternative treatment option in regions where commercial incentives are either difficult to obtain or unavailable.

The study of the distribution of pulmonary ventilation of the MPIS compared to another branded IS, with objective and state-of-the-art measures, will complement and allow the objective identification of the effectiveness of the MPIS and promote spaces for discussion and critical reflection on the reasons that condition its implementation.

The EIT is a diagnostic tool that uses the electrical properties of tissue to provide information in a non-invasive, continuous, bedside, radiation-free manner, and the use of this monitoring tool will help validate the MPIS in terms of lung ventilation distribution [10,18].

It is essential to emphasize that, given the pilot nature of this study, the results obtained should be limited. However, the results may serve as a basis for subsequent trials, particularly in individuals with thoracic anomalies leading to lung limitation.

In summary, the implementation of this protocol will enable the comparison of pulmonary ventilation between two incentive spirometers. The findings have the potential to support the utilization of a low-cost device, thus promoting greater accessibility for populations that require it.

## Supporting information

**S1 File. SPIRIT checklist.**
(PDF)

**S2 File. Full-length study protocol (English).**
(PDF)

**S3 File. Appendix table.** World Health Organization trial registration data set.
(PDF)

## Author Contributions

**Conceptualization:** Noraelena Mera, Esther Cecilia Wilches Luna, Vicente Benavides-Cordoba.

**Data curation:** Noraelena Mera, Esther Cecilia Wilches Luna, Vicente Benavides-Cordoba.

**Formal analysis:** Noraelena Mera, Esther Cecilia Wilches Luna, Vicente Benavides-Cordoba.

**Investigation:** Noraelena Mera, Esther Cecilia Wilches Luna, Vicente Benavides-Cordoba.

**Methodology:** Noraelena Mera, Esther Cecilia Wilches Luna, Vicente Benavides-Cordoba.

**Project administration:** Noraelena Mera, Esther Cecilia Wilches Luna, Vicente Benavides-Cordoba.

**Resources:** Noraelena Mera.

**Supervision:** Esther Cecilia Wilches Luna.

**Validation:** Vicente Benavides-Cordoba.

**Visualization:** Noraelena Mera, Esther Cecilia Wilches Luna, Vicente Benavides-Cordoba.

**Writing – original draft:** Noraelena Mera, Esther Cecilia Wilches Luna, Vicente Benavides-Cordoba.

**Writing – review & editing:** Noraelena Mera, Esther Cecilia Wilches Luna, Vicente Benavides-Cordoba.

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
