## [Decision Letter · Decision Letter 0]

15 Feb 2024

PONE-D-23-39727Comparative analysis of pulmonary ventilation distribution between low-cost and branded incentive spirometers using electrical impedance tomography in healthy adults: Study protocol.PLOS ONE

Dear Dr. Benavides-Cordoba,

Thank you for submitting your manuscript to PLOS ONE. After careful consideration, we feel that it has merit but does not fully meet PLOS ONE’s publication criteria as it currently stands. Therefore, we invite you to submit a revised version of the manuscript that addresses the points raised during the review process.

We look forward to receiving your revised manuscript.

Kind regards,

Davor Plavec, MD, MSc, PhD, Prof.

Academic Editor

PLOS ONE

Journal Requirements:

**Additional Editor Comments:**

Dear Authors,

please make a revision of your manuscript as suggested by the reviewers or write a detailed rebuttal on a point-by-point basis.

Reviewers' comments:

Reviewer's Responses to Questions

**Comments to the Author**

1. Does the manuscript provide a valid rationale for the proposed study, with clearly identified and justified research questions?

Reviewer #1: Yes

Reviewer #2: Partly

Reviewer #3: No

2. Is the protocol technically sound and planned in a manner that will lead to a meaningful outcome and allow testing the stated hypotheses?

Reviewer #1: Yes

Reviewer #2: No

Reviewer #3: No

3. Is the methodology feasible and described in sufficient detail to allow the work to be replicable?

Reviewer #1: Yes

Reviewer #2: No

Reviewer #3: No

4. Have the authors described where all data underlying the findings will be made available when the study is complete?

Reviewer #1: Yes

Reviewer #2: Yes

Reviewer #3: Yes

5. Is the manuscript presented in an intelligible fashion and written in standard English?

Reviewer #1: No

Reviewer #2: No

Reviewer #3: No

6. Review Comments to the Author

You may also provide optional suggestions and comments to authors that they might find helpful in planning their study.

Reviewer #1: English language must be improved. Some parts of the sentences are repeating. There are so many grammatical mistakes. Some sentences aren’t understandable. Manuscript needs detailed English polishing and, in this format, it can’t be published.

Tidal Variation – short is TV not VT as it is written in study protocol. Please change accordingly.

Minute tidal variation – short is MTV no MVT as it is written in study protocol. Please change accordingly.

When you read manuscript draft it’s not clear will you use the same branded IS or different ones - please specify in text of PONE-D-23-39727_reviewer. In Spirit 2013 Checklist type of IS is specified.

Reviewer #2: This is a study protocol corresponding to a cross-over trial to compare the measurement of the distribution of pulmonary ventilation of commercial flow IS versus the low-cost MPIS. The trial is registered within clinicaltrials.gov with a specific NCT number, and has been approved by the concerned IRB/Ethics Committee. The study objectives are on target, and the study design looks adequate. However, I mostly have some concerns/comments in the statistical design and analytical framework, and CONSORT guidelines, which may require attention:

1. Methods:

Methods reporting need some work. An orderly manner is suggested, following CONSORT guidelines, without repeating information, such as Trial Design, Participant Eligibility Criteria and settings, Interventions, Outcomes, sample size/power considerations, Interim analysis and stopping rules, Randomization (details on random number generation, allocation concealment, implementation), Blinding issues, etc, should be mentioned. The authors are advised to create separate subsections for each of the possible topics (whichever necessary), and that way produce a very clear writeup. I see there is already a sincere effort. They are advised to write it carefully, following nice examples in the manuscript below:

https://www.sciencedirect.com/science/article/pii/S0889540619300010

Specific comments:

(a) For instance, the randomization and allocation concealment should be made very clear (they are NOT the same thing); the trial staff recruiting patients should NOT have the randomization list. Randomization should be prepared by the trial statistician, and he/she would not participate in the recruiting.

(b) Details on the randomization, with reference to the crossover design, is needed; just stating that randomization.com will be used to generate is not enough.

(c) I was surprized to find no justification of sample size/power in the draft; just stating that 30 subjects (15 and 15) will be recruited is a missed opportunity. There exists suggestions for constructing such numbers for pilot randomized cross-over trials: https://www.ncbi.nlm.nih.gov/pmc/articles/PMC4876429/

Sample sizes need to be presented using the primary endpoint, with mention of the desired effect sizes, and appropriate statistical tests.

2. Writing style: The writing style needs a serious facelift. Certain areas have misconstructed sentences.

3. Discussion Section: Given the study was designed based on a pilot study, the Discussion section should clearly state that the findings of this study would be only limited to the enrolled sample, and should allude to future trials with larger sample sizes (under medium to large effect sizes), and covering other populations.

Reviewer #3: Dear Sir,

First, the article needs some serious grammar, spelling, and incomplete sentences revision. But what is more important, as I understand, you will be testing the healthy volunteers. If I am not wrong, in those persons the results of EIT should be normal independently of the device used so no difference will be default result and the study can not fulfill its primary purpose of comparing the efficiency of two different devices. I would suggest comparing the effects of two different devices in the population with the respiratory problems consistent with the intended use (e.g. after thoracic surgery) in the same crossover manner but with the shorter period between the tests.

Regards

7. PLOS authors have the option to publish the peer review history of their article (what does this mean?). If published, this will include your full peer review and any attached files.

Reviewer #1: No

Reviewer #2: No

Reviewer #3: No

---

## [Author Response · Author response to Decision Letter 0]

13 Mar 2024

March, 2024

Respected

EDITORIAL BOARD AND REVIEWERS

PLOS ONE

Kind regards.

Through this letter, the authors would like to confirm the receipt of the observations made by the reviewers of the article titled "Comparative analysis of pulmonary ventilation distribution between low-cost and branded incentive spirometers using electrical impedance tomography in healthy adults: Study protocol," which has been submitted for consideration in PLOS One. 

We want to express our sincere gratitude for the valuable observations provided by each of the three reviewers. We believe that these observations are highly relevant and will significantly contribute to improving the quality of the manuscript, ensuring that it fulfills its objective of reporting the protocol of this clinical trial. 

We are attaching three main files: the current one, which is the response to the reviewers, the revised manuscript in its clean version, and the revised manuscript with track changes.

Below, we will proceed to respond in detail to each of the observations received.

REVIEWER #1: 

• English language must be improved. Some parts of the sentences are repeating. There are so many grammatical mistakes. Some sentences aren’t understandable. Manuscript needs detailed English polishing and, in this format, it can’t be published.

The authors are grateful for your feedback. We have enhanced the English language writing throughout the entire text. Furthermore, a Native North American language teacher has reviewed and edited the language for accuracy and clarity.

• Tidal Variation – short is TV not VT as it is written in study protocol. Please change accordingly.

Thank you for your observation; the corresponding adjustment has been made.

• Minute tidal variation – short is MTV no MVT as it is written in study protocol. Please change accordingly.

Thank you for your observation; the corresponding adjustment has been made.

• When you read manuscript draft it’s not clear will you use the same branded IS or different ones - please specify in text of PONE-D-23-39727_reviewer. In Spirit 2013 Checklist type of IS is specified.

Thank you very much for your observation. In section 2.6.1, the description of interventions includes the Triflo II and MPIS devices.

REVIEWER #2 

This is a study protocol corresponding to a cross-over trial to compare the measurement of the distribution of pulmonary ventilation of commercial flow IS versus the low-cost MPIS. The trial is registered within clinicaltrials.gov with a specific NCT number, and has been approved by the concerned IRB/Ethics Committee. The study objectives are on target, and the study design looks adequate. However, I mostly have some concerns/comments in the statistical design and analytical framework, and CONSORT guidelines, which may require attention:

Methods:

• Methods reporting need some work. An orderly manner is suggested, following CONSORT guidelines, without repeating information, such as Trial Design, Participant Eligibility Criteria and settings, Interventions, Outcomes, sample size/power considerations, Interim analysis and stopping rules, Randomization (details on random number generation, allocation concealment, implementation), Blinding issues, etc, should be mentioned. The authors are advised to create separate subsections for each of the possible topics (whichever necessary), and that way produce a very clear writeup. I see there is already a sincere effort. They are advised to write it carefully, following nice examples in the manuscript below: https://www.sciencedirect.com/science/article/pii/S0889540619300010

The authors appreciate the observations made. We have completely restructured the methodology in terms of the distribution of each of its subchapters. This reorganization was based on the recommendations of the evaluator, the suggested article, and other protocols published in PLOS One.

Specific comments:

• For instance, the randomization and allocation concealment should be made very clear (they are NOT the same thing); the trial staff recruiting patients should NOT have the randomization list. Randomization should be prepared by the trial statistician, and he/she would not participate in the recruiting.

The researchers acknowledge and comply with the recommendations made by the evaluator. It should be noted that the randomization process will be conducted by a statistician who is not part of the research team. We have provided a more detailed description of the allocation protocol.

• (b) Details on the randomization, with reference to the crossover design, is needed; just stating that randomization.com will be used to generate is not enough.

We appreciate your observations. The process has been described in more detail.

• (c) I was surprized to find no justification of sample size/power in the draft; just stating that 30 subjects (15 and 15) will be recruited is a missed opportunity. There exists suggestions for constructing such numbers for pilot randomized cross-over trials: https://www.ncbi.nlm.nih.gov/pmc/articles/PMC4876429/. Sample sizes need to be presented using the primary endpoint, with mention of the desired effect sizes, and appropriate statistical tests.

The authors appreciate and agree with your observation. Although the sample calculation remained unchanged, we have provided a thorough explanation to justify this calculation, focusing on the type of design we are implementing.

• 2. Writing style: The writing style needs a serious facelift. Certain areas have misconstructed sentences.

The authors are grateful for your feedback. We have enhanced the English language writing throughout the entire text. Furthermore, a Native North American language teacher has reviewed and edited the language for accuracy and clarity.

• 3. Discussion Section: Given the study was designed based on a pilot study, the Discussion section should clearly state that the findings of this study would be only limited to the enrolled sample, and should allude to future trials with larger sample sizes (under medium to large effect sizes), and covering other populations.

The authors acknowledge your recommendation and have made adjustments to the discussion based on the reviewer's suggestions.

REVIEWER #3: 

• Dear Sir. First, the article needs some serious grammar, spelling, and incomplete sentences revision. But what is more important, as I understand, you will be testing the healthy volunteers. If I am not wrong, in those persons the results of EIT should be normal independently of the device used so no difference will be default result and the study can not fulfill its primary purpose of comparing the efficiency of two different devices. I would suggest comparing the effects of two different devices in the population with the respiratory problems consistent with the intended use (e.g. after thoracic surgery) in the same crossover manner but with the shorter period between the tests.

The authors appreciate the recommendations and have made significant efforts to improve the entire document in terms of writing and composition. The paper was reviewed by a native North American language professor, who also edited the document to enhance its clarity and comprehension.

Regarding the target population of the protocol, the authors consider it an important recommendation, which we will apply for the design of a study involving a population with restrictive lung impairment, aiming to assess effectiveness. However, for this protocol, we continue to propose a healthy population as we intend to evaluate the device's action in a proof-of-concept phase. Hence, the selected population and sample size remain unchanged.

With these adjustments, the authors hope to meet the requirements that will allow our article to be accepted in such a prestigious journal. We will remain attentive to any requirements that may arise.

---

## [Decision Letter · Decision Letter 1]

5 Apr 2024

Comparative analysis of pulmonary ventilation distribution between low-cost and branded incentive spirometers using electrical impedance tomography in healthy adults: Study protocol.

PONE-D-23-39727R1

Dear Dr. Benavides-Cordoba,

We’re pleased to inform you that your manuscript has been judged scientifically suitable for publication and will be formally accepted for publication once it meets all outstanding technical requirements.

Kind regards,

Davor Plavec, MD, MSc, PhD, Prof.

Academic Editor

PLOS ONE

Additional Editor Comments (optional):

Dear Authors,

your manuscript is now acceptable for publication.

Reviewers' comments:

Reviewer's Responses to Questions

**Comments to the Author**

1. Does the manuscript provide a valid rationale for the proposed study, with clearly identified and justified research questions?

Reviewer #1: Yes

Reviewer #2: Partly

2. Is the protocol technically sound and planned in a manner that will lead to a meaningful outcome and allow testing the stated hypotheses?

Reviewer #1: Yes

Reviewer #2: Partly

3. Is the methodology feasible and described in sufficient detail to allow the work to be replicable?

Reviewer #1: Yes

Reviewer #2: Yes

4. Have the authors described where all data underlying the findings will be made available when the study is complete?

Reviewer #1: Yes

Reviewer #2: Yes

5. Is the manuscript presented in an intelligible fashion and written in standard English?

Reviewer #1: Yes

Reviewer #2: No

6. Review Comments to the Author

You may also provide optional suggestions and comments to authors that they might find helpful in planning their study.

Reviewer #1: In abstract in front of EIT full name should be written (Electrical Impedance Tomography ).

Authors have implemented all suggested changes.

English language has been improved.

Reviewer #2: The authors addressed my previous round of comments with significant satisfaction. I have no further comments.

7. PLOS authors have the option to publish the peer review history of their article (what does this mean?). If published, this will include your full peer review and any attached files.

Reviewer #1: No

Reviewer #2: No

---

## [Editor Report · Acceptance letter]

26 Apr 2024

PONE-D-23-39727R1 

PLOS ONE

Dear Dr. Benavides-Cordoba, 

I'm pleased to inform you that your manuscript has been deemed suitable for publication in PLOS ONE. Congratulations! Your manuscript is now being handed over to our production team.

Kind regards, 

on behalf of

Dr. Davor Plavec 

Academic Editor

PLOS ONE